# Can the Adjustment of China’s Grain Purchase and Storage Policy Improve Its Green Productivity?

**DOI:** 10.3390/ijerph19106310

**Published:** 2022-05-23

**Authors:** Jingdong Li, Qingning Lin

**Affiliations:** 1Institute of Geographic Sciences and Natural Resources Research, Chinese Academy of Sciences, Beijing 100101, China; lijingdong@igsnrr.ac.cn; 2Key Laboratory of Regional Sustainable Development Modeling, Chinese Academy of Sciences, Beijing 100101, China; 3Institute of Agricultural Economics and Development, Chinese Academy of Agricultural Sciences, Beijing 100081, China

**Keywords:** DID model, green productivity, maize production, purchase and storage policy

## Abstract

While the sustainability of grain production has been extensively studied, there have been few studies focusing on the impact of grain policy adjustment on its sustainable production, and the quantitative relationship between these two aspects and the internal mechanism is not completely clear. The main objective of this paper was to explore the impact of grain purchase and storage policy (GPSP) adjustment on its green productivity by expounding the evolution logic and influence mechanism of GPSP. Therefore, taking maize production as an example, this paper constructs the analysis framework of the evolution logic and influence mechanism, and the super-epsilon-based measure model (Super-EBM) is adopted to measure maize green productivity (MGP) in main producing areas from 1997 to 2019, then two groups of difference-in-differences (DID) models are constructed to study the influence of the temporary purchase and storage policy (TPSP) and the producer subsidy policy (PSP) on MGP. The main conclusions include: the implementation of TPSP reduces MGP in Heilongjiang, Jilin, Liaoning and Inner Mongolia (experimental group), whereas the implementation of PSP improves MGP in these provinces is due to the difference in policy effects under the different regulatory objectives and measures; under the demonstration effect of two policies, the increase in effective irrigation and agricultural financial expenditure are important factors to improve MGP, but the backwardness of agricultural mechanization has been hindering the improvement of MGP; after the reform from TPSP to PSP, the continuous increase in production capacity hinders the improvement of MGP under the support effect, the impacts of farmers’ income and agricultural production price on MGP both shift from negative to positive under the wealth effect, and the influence of production agglomeration on MGP shifts from negative to positive under the siphon effect. The excessive implementation of GPSP has seriously affected the sustainability of grain production, thus, this study has certain practical significance and guiding value. The paper emphasizes that the effective way to achieve sustainable food production is to combine the adjustment of GPSP with improving the subsidy mechanism, enhancing the agricultural mechanization and maintaining the appropriate scale of operation.

## 1. Introduction

Food security is the cornerstone of national economic development and social stability. It is a complex and systematic problem, which involves not only the links of food production, supply and consumption, but is also closely related to the links of food storage and transportation (such as warehousing and logistics). Nevertheless, grain production remains the most fundamental factor to guarantee food security. Both developing and developed countries attach great importance to grain self-sufficiency [1]. In the past ten years, the implementation of national grain purchase and storage policy and other grain policies played a positive role in increasing grain output and stabilizing grain market [2,3]. However, excessive implementation of national food policies will not only lead to market distortion [4], but also lead to excessive consumption of agricultural resources and deterioration of ecologic environment due to obsessive pursuit of grain output. The Sustainable Development Goals (SDGs) of the United Nations have made clear provisions for hunger eradication, food security and sustainable agricultural development, to achieve sustainable security of food supply (Goal 2). The latest definition of food security includes six aspects: availability, access, utilization, stability, agency and sustainability [5]. Among these factors, how to maintain the sustainable development of grain production and the role of agencies (especially governmental departments) in this have become a hot topic.

Through continuous adjustment and reform of relevant agricultural policies, government departments have ensured food security and increased farmers’ income, to match the level of agricultural production with social and economic development, and focus on sustainable high-quality development [6]. Some studies have concluded that the implementation of agricultural policies has significantly improved farmers’ planting willingness and planting area, which is conducive to optimizing the utilization efficiency of production elements, thus improving the yield and productivity [7,8]. However, some studies believe that the burdensome agricultural policies lead to the excessive use of chemical fertilizers in agricultural production, which increase greenhouse gas emissions and non-point source pollution [9], thus aggravating the pressure of agricultural ecological environment and challenging the sustainability of agricultural production [10]. Focusing on the research of grain policies on sustainable production, some studies believe that price-support policies, agricultural insurance systems, market-aid policies, preferential loan policies, and other agricultural transfer payment policies are import in improving grain productivity, reducing chemical fertilizers input and protecting the agricultural ecological environment [11,12,13,14,15]. However, some studies believe that agricultural policies such as temporary storage and subsidies are inefficient [16]. For example, government transfer payments reduce the productivity of Canadian wheat [17]; crop subsidies and environmental programs of the European Union have an inhibitory effect on crop production efficiency [18,19], and also bring about serious resource consumption and ecological environment degradation [20,21], thereby affect grain productivity and sustainable development [22].

Agricultural productivity is widely used as an effective evaluation index when measuring agricultural production level. Data Envelopment Analysis (DEA) and Solow residual value and Stochastic Frontier Analysis (SFA) are gradually become the common methods for measuring agricultural productivity [23,24,25,26]. In recent years, issues such as agricultural environmental pollution, excessive input of chemical fertilizers and pesticides have attracted extensive attention in the academic community, while traditional SFA and DEA methods do not take the environmental pollution into account. Therefore, in recent years, many studies have included environmental pollution as an undesirable output into the agricultural production process to measure agricultural green productivity under environmental constraints [27]. Meanwhile, the influencing factors of agricultural green productivity are also the focus, and relevant studies discuss the influence of technology level [28,29], agricultural industrial structure [29,30], government intervention [31], climate change and other factors [26]. However, few studies focus on the influence of agricultural policy adjustments on agricultural green productivity.

As the most populous country in the world, China’s food security has attracted global attention. In recent years, the pressure of continuous population growth, limited land resources, greenhouse gas emissions and environmental damage have brought great challenges to the sustainability of food security. How to achieve sustainable development of food production has become a key concern of the Chinese government. In 2021, under the goals of carbon peak and carbon neutralization, the Chinese government issued the 14th Five-year Plan for Green Agricultural Development [32], which aims to control the input of chemical fertilizers, pesticides and other pollution sources, strengthen the protection of agricultural resources, improve the utilization efficiency of various input elements in grain production and reduce the pressure on the ecological environment, and the sustainability of grain production can be realized through the path of improving the grain green productivity. China’s grain policy, especially the grain purchase and storage policy, has made great contributions to ensuring food security, but its impact on the sustainability of grain production needs to be studied according to specific policy items. As one of the important grains, maize has the characteristics of diversified uses, and has a substitution relationship with wheat, rice, barley and other grain varieties in production and consumption. Meanwhile, the implementation of the temporary and storage policies in recent years have significantly increased grain output and stably increased farmers’ income, but also have brought some negative effects such as overcapacity of maize, heavy financial burden, waste of resources and environmental pollution. Due to the increasing acuteness of these problems, the Chinese government cancelled the temporary purchase and storage policy in 2016, and reformed it into a producer subsidy policy, in order to adjust the grain structure and optimize the allocation of resources. Therefore, this paper takes the reform of maize purchase and storage policy as an example to explore the influence of grain policies on the sustainability of grain production, which is representative and typical.

This study contributes to existing literature by constructing a theoretical framework for analyzing the effects of grain purchase and storage policy, using the advanced Super-Epsilon Based Measure model (Super-EBM) on the basis of considering the undesirable output to measure the maize green productivity (MGP), and conducting natural experiments based on the implementation of the temporary purchase and storage policy (TPSP) and the implementation of producer subsidy policy (PSP). DID method is applied to explore the different impact of various policies on MGP, in order to provide a practical basis for the green development of China’s grain industry and provide ideas for the adjustment of the purchase and storage policies. The study is structured as follows: firstly, it expounds the evolutionary logic of maize purchase and storage policy, and constructs the framework for analyzing the influencing factors and policy effects of grain green productivity; secondly, Super-EBM model is used to calculate MGP in each province based on the data of production elements in maize producing areas; finally, two groups of DID models are established to study the impacts of TPSP and PSP on MGP, respectively.

## 2. Analysis of Policy Evolution and Influence Mechanism

### 2.1. Evolutionary Logic of Maize Purchase and Storage Policy

Grain purchase and storage policy is the center of China’s grain security policies, which has made great contributions to increasing grain output, stabilizing grain price, increasing farmers’ income and ensuring food security. The policy has been continuously adjusted and reformed in different historical backgrounds, according to the overall goal of agricultural policy, to meet the needs of economic and social development [33]. From 1997 to 2019, the maize purchase and storage policies experienced three stages: protective price policy (1997–2007), temporary purchase and storage policy (2008–2015) and producer subsidy policy (2016–2019). The three policies are shown in detail in Table 1.

(1)Protective price policy (PPP): The government formulates the protective price before the grain put on the market. If the market price is higher than the protective price after grain put on the market, the grain purchase and storage enterprises (state-owned) will make monopolistic purchase of grain at the market price; otherwise, they will purchase grain at the protective price.(2)Temporary purchase and storage policy (TPSP): The central government announces the maize temporary purchase and storage price during the harvest. State-owned grain purchase and storage enterprises and private enterprises entrusted by the government purchase maize in Heilongjiang, Jilin, Liaoning and Inner Mongolia at this price.(3)Producer subsidy policy (PSP): The price of maize in Heilongjiang, Jilin, Liaoning and Inner Mongolia is completely determined by the supply and demand after they put on the market. Producers will get subsidy when the market price is low, and the amount of subsidy is determined by the central government, based on the overall consideration of previous production cost, current supply and demand level and basic profit.

### 2.2. Influence Mechanism and Policy Effects

The factors affecting agricultural green productivity mainly include agricultural production condition, farmers’ production decision, regional agglomeration capacity, government financial support, economic development level and natural disaster impact (Figure 1). The source power affecting agricultural productivity is the basic conditions of agricultural production. When the basic conditions match the level of productivity, this can improve the utilization efficiency of production means, improve marginal desirable output and reduce the undesirable output [40,41,42]. The farmers’ production decisions affect agricultural green productivity in that, on the one hand, it affects the productivity by changing the amount and proportion of the production elements [43,44]; and on the other hand, it has a scale effect on production by adjusting the production scale [45]. Regional agglomeration capacity affects agricultural green productivity in that, on the one hand, it can promote technology and knowledge spillover, encourage the development and application of energy-saving technology to improve resource utilization efficiency [46]; on the other hand, it can increase energy efficiency by improving energy quality and optimizing energy structure [47]. Government’s financial expenditure affects agricultural green productivity in that, on the one hand, improves agricultural productivity by increasing agricultural machinery input and utilization rate of improved varieties, and improving management level and the quality of socialized services [30,31,44]; on the other hand, it also affects the green productivity by changing the profit expectations of stakeholders and the status of market risks [48]. With the economic and social development, producers are more aware of the energy conservation and environmental protection, and are more willing to accept and adopt green technology [42,49]. Moreover, it affects green production technology by affecting income levels and business decisions [50]. Natural disaster is the main factor of instability of grain production. The most significant direct impact of natural disasters is the decline in grain yield. To reduce the losses caused by this decline, producers often adopt multiple cropping, multiple tillage, flood drainage, and excessive application of chemical fertilizers and pesticides to ensure output [51,52]. The repeated or excessive use of input elements leads to low productivity and waste of resources.

The effects of purchase and storage policy on grain green productivity mainly include support effect, wealth effect, demonstration effect and siphon effect (as shown in Figure 1).

(1)Support effect. The implementation of the purchase and storage policy provides basic struts for grain production and food security. Purchase and storage policy featuring the minimum price purchase policy (MPPP) and TPSP, in combination with agricultural subsidies, finance, science and technology supporting policies, has increased grain output steadily especially since 2003, and food supply security has been guaranteed [38]. Despite the increase in grain output, the impact on grain productivity is uncertain. Problems caused by the purchase and storage policy, such as waste of resources, overcapacity and price inversion [35,39], which hinder the improvement of green productivity.(2)Wealth effect. The grain purchase and storage policy ensure price stability and purchase channels, which stabilizes farmers’ income, reduces market risks, and improves their enthusiasm of grain production [38]. It achieves rapid growth of grain production by encouraging farmers to expand production scale or increase inputs. However, the expansion of production scale is often confronted with the constraints of land fragmentation, the rise of production costs, natural disasters and increase in market operation risks, resulting in the loss of productivity [37,53]. Moreover, the unreasonable growth of agricultural inputs can easily lead to overcapacity, resource mismatch and decline in marginal output [39], which reduces grain productivity.(3)Demonstration effect. The grain purchase and storage policy have strong guidance [54,55]. Stable implementation of the policy will help to encourage local governments and relevant departments to attach greater importance to agricultural production, increase financial expenditure on agriculture, and improve production infrastructure. It can also increase the scientific and educational departments’ investment in research and development and talent cultivation. Furthermore, it increases the financial support from banks and credit institutions, and improves the efficiency of environmental testing by environment protection departments [51,56]. Therefore, it promotes the improvement of grain green productivity.(4)Siphon effect. In the provinces that implement grain purchase and storage policy, advantageous conditions are gradually formed, with higher grain purchase price, stable purchase channels, sound agricultural infrastructure, and higher productivity and convenient socialized services, which results in the agglomeration of grain production. Moreover, the existing advantageous conditions will also attract the capital, technology, labor and other production elements in the surrounding areas to flow to the local region, and forming a siphon effect [46], so as to promote the development of green productivity in the local region.

## 3. Materials and Methods

### 3.1. Epsilon-Based Measure (EBM) Model

Stochastic frontier analysis (SFA) and data envelopment analysis (DEA) are the main methods to measure green productivity [57]. Compared with the SFA method, DEA, as a non-parametric method, does not rely on the functional relationship between input and output, and is suitable for measuring the efficiency of complex systems which have multiple inputs and multiple outputs [58,59]. Therefore, DEA is selected in this study.

The influence of non-radial relaxation variables should be fully considered, to achieve compatibility of radial and non-radial relaxation variables, and guarantee the original information of the frontier projected value of efficiency. However, the traditional DEA method cannot meet the requirements after including the undesirable output. The epsilon-based measure (EBM) model proposed by Tone and Tsutsutsui [60] under the DEA method can better solve these problems. Moreover, in order to compare the differences between decision-making units (DMUs) with the same efficiency value of 1, this study follows the method of Wu et al. [47] and Zhao et al. [61] and adopts the super-EBM model to measure MGP. At present, the super-EBM model has been widely used in the research of green economic efficiency, green growth efficiency and green innovation efficiency [47,62,63].

The super-EBM model with undesirable output can be expressed as:(1)$$r^{*}=\min\left[\frac{\theta -\varepsilon_{x}\sum_{i=1}^{m}\frac{w_{i}^{-}s_{i}^{-}}{x_{ik}}}{\varphi +\varepsilon_{y}\sum_{r=1}^{q}\frac{w_{r}^{g+}s_{r}^{g+}}{y_{rk}}+\varepsilon_{v}\sum_{t=1}^{p}\frac{w_{t}^{b-}s_{t}^{b-}}{v_{tk}}}\right]$$
(2)$$\begin{array}{cccc} \sum_{j=1,j\neq k}^{n}x_{ij}\lambda_{j}-s_{i}^{-}\leq \theta \cdot x_{ik}, & i=1,\ldots ,m, & \lambda \geq 0, & s^{-}\geq 0 \end{array}$$
(3)$$\begin{array}{cccc} \sum_{j=1,j\neq k}^{n}y_{rj}\lambda_{j}-s_{r}^{g+}\geq \varphi \cdot y_{rk}, & r=1,\ldots ,q, & \lambda \geq 0, & s^{g+}\geq 0 \end{array}$$
(4)$$\begin{array}{cccc} \sum_{j=1,j\neq k}^{n}v_{tj}\lambda_{j}-s_{t}^{b-}\leq v_{tk}, & t=1,\ldots ,p, & \lambda \geq 0, & S^{b-}\geq 0 \end{array}$$
(5)$$\begin{array}{ccc} \sum_{i=1}^{m}w_{i}^{-}=1\left(w_{i}^{-}\geq 0\right), & \sum_{i=1}^{m}w_{r}^{g+}=1\left(w_{r}^{g+}\geq 0\right), & \sum_{i=1}^{m}w_{t}^{b-}=1\left(w_{t}^{b-}\geq 0\right) \end{array}$$
where $r^{*}$ represents the MGP under the super-EBM model; $x_{ij}$ represents the input variable matrix of maize, and the specific indicators include planting area, fertilizer, pesticide, agricultural film, diesel oil, seed, electricity for irrigation, labor and machinery in maize production [28,64]; $y_{rj}$ is the desirable output, which is expressed in maize output; $v_{tj}$ is the undesirable output, which is expressed by the sum of carbon emissions and non-point source pollution based on Chen et al. [64]; $s_{i}^{-}$, $s_{r}^{g+}$ and $s_{t}^{b-}$ represent slacks of inputs, slacks of desirable outputs and slacks of undesirable outputs, respectively; $w_{i}^{-}$, $w_{r}^{g+}$ and $w_{t}^{b-}$ represent the relative importance of various input indicators, desirable outputs and undesirable outputs; θ is the efficiency value under input orientation; φ is the efficiency value under output orientation; ε represents the importance of the non-radial part; and $\varepsilon \in \left[\begin{array}{cc} 0, & 1 \end{array}\right]$.

### 3.2. Difference-in-Differences (DID) Model

When evaluating the effect of policy implementation, the difference-in-differences (DID) method based on natural experiment is a typical research tool, which can also effectively reduce the endogenous problems between variables, and thus it has been widely used in the fields of natural sciences and social sciences (such as biological science, medicine, economics, sociology, law and management) [65,66,67,68]. Natural experimental method needs to divide the sample provinces into experimental group and control group to study policy effect. The experimental group is the areas where the policy is implemented, and the control group is the area where the policy is not implemented. Moreover, the sample period is divided into pre-implementation period and implementation period, according to the time of policy implementation. This paper mainly investigates the influence of the reform of maize purchase and storage policy on MGP. Sample provinces are divided into experimental group and control group according to the implementation of TPSP and PSP. Research period is divided into three stages according to the time of implementation: the period before TPSP (the period of PPP) (1997–2007), period of TPSP (2008–2015) and the period of reform of TPSP (the period of PSP) (2016–2019). Among these, experimental group include Heilongjiang, Jilin, Liaoning and Inner Mongolia, which implemented TPSP in 2008, abolished TPSP in 2016 and reformed it into PSP. The other provinces that did not implement TPSP or PSP are the control group. Based on the analysis above, this study establishes two groups of DID models to investigate the influence of the implementation of TPSP and PSP on MGP. Time of the sample is distinguished to eliminate the mixed influence among PPP, TPSP and PSP, and to enhance the stability of the estimation results of DID models. The sample period of the natural experiment of TPSP is 1997–2015, and that of PSP is 2008–2019.

Firstly, the DID model to analyze the influence of the TPSP implementation on MGP can be expressed as:(6)$$MGP_{it}^{TPSP}=\alpha^{TPSP}+\beta^{TPSP}DID_{i}^{TPSP}\times year_{t}^{TPSP}+\rho^{TPSP}X_{it}+\nu_{it}^{TPSP}+\eta_{it}^{TPSP}+\varepsilon_{it}^{TPSP}$$
where *MGP_it_^TPSP^* represents the MGP in the natural experiment of TPSP, *i* is province, *t* represents year; *DID_i_^TPSP^* is the dummy variable of the experience group, if province *i* belongs to the experience group *DID_i_^TPSP^* = 1, otherwise *DID_i_^TPSP^* = 0; *year_i_^TPSP^* is a dummy variable of year, and the experience group implemented the TPSP in 2008, so *year_i_^TPSP^* = 0 from 1997 to 2007 and *year_i_^TPSP^* = 1 from 2008 to 2015; *DID_i_^TPSP^* × *year_i_^TPSP^* is the policy variable, and its coefficient $\beta^{TPSP}$ represents the degree and direction of the TPSP’s influence on MGP; *X_it_* represents a series of control variables; $\nu_{it}^{TPSP}$ and $\eta_{it}^{TPSP}$ are province-fixed effects and year-fixed effects, respectively; $\varepsilon_{it}^{TPSP}$ represents the random errors.

Secondly, the DID model to analyze the influence of the PSP implementation on MGP can be expressed as:(7)$$MGP_{it}^{PSP}=\alpha^{PSP}+\beta^{PSP}DID_{it}^{PSP}\times year_{t}^{PSP}+\rho^{PSP}X_{it}+\nu_{it}^{PSP}+\eta_{it}^{PSP}+\varepsilon_{it}^{PSP}$$
where *MGP_it_^PSP^* represents the MGP in the natural experiment of PSP, *i* is province, *t* represents year; *DID_i_^PSP^* is the dummy variable of the experience group, if province *i* belongs to the experience group *DID_i_^PSP^* = 1, otherwise *DID_i_^PSP^* = 0; *year_i_^PSP^* is a dummy variable of year, and the experience group implemented the TPSP in 2016, so *year_i_^PSP^* = 0 from 2008 to 2015 and *year_i_^PSP^* = 1 from 2016 to 2019; *DID_i_^PSP^* × *year_i_^PSP^* is the policy variable, and its coefficient $\beta^{PSP}$ represents the degree and direction of the PSP’s influence on MGP; *X_it_* represents a series of control variables; $\nu_{it}^{PSP}$ and $\eta_{it}^{PSP}$ are province-fixed effects and year-fixed effects, respectively; $\varepsilon_{it}^{PSP}$ represents the random errors.

### 3.3. Variable Selection and Data Source

Based on the influence mechanism of MGP and the policy effects of grain purchase and storage policy, this paper selects the agricultural machinery intensity (*AMI*) and effective irrigated rate (*EIR*) to reflect the agricultural production condition; selects the grain production capacity (*GPC*) and agricultural production price index (*API*) to reflect the Farmers’ production decisions, selects production agglomeration (*PA*) as the proxy variables of regional agglomeration capacity, selects agricultural financial expenditure rate (*AFE*) to reflect the government financial support, selects rural income inequality (*RII*) and urbanization rate (*UR*) to reflect the economic development level, selects disaster incidence (*DI*) as the proxy variable of natural disaster impact, and considering the increasing impact of climate change on food production in recent years [69], this paper adds temperature fluctuation, precipitation fluctuation and sunshine change as the characteristic vector of climate conditions.

Agricultural machinery intensity (*AMI*):*AMI_it_ = TPAM_it_*/*TPA_it_*(8)
where *TPAM_it_* represents the total power of agricultural machinery of province *i* in year *t*; *TPA_it_* represents the total planting area.

Effective irrigated rate (*EIR*):*EIR_it_ = EIA_it_*/*TCA_it_*(9)
where *EIA_it_* represents the effective irrigated area; *TCA_it_* represents the total cultivated area.

Grain production capacity (*GPC*):*GPC_it_ = GP_it_*/*AP_it_*(10)
where *GP_it_* represents the staple grain production (include maize, wheat and rice); *AP_it_* represents the labor input for staple grain production.

Production agglomeration (*PA*):*PA_it_ =* (*MP_it_*/*GP_it_*)/(*NMP_t_*/*NGP_t_*)(11)
where *MP_it_* represents the maize production by province; *GP_it_* represents the staple grain production (include maize, wheat and rice) by province; *NMP_t_* represents the national maize production; *NGP_t_* represents the national staple grain production.

Agricultural financial expenditure rate (*AFE*):*AFE_it_ = AFE_it_*/*LFE_it_*(12)
where *AFE_it_* represents the local agricultural fiscal expenditure; *LFE_it_* represents the local fiscal expenditure.

Rural income inequality (*RII*):*RII_it_ = PRR_it_/PUR_it_*(13)
where *PRR_it_* represents the per capita income of rural residents; *PUR_it_* represents the per capita income of urban residents.

Urbanization rate (*UR*):*UR_it_ = UP_it_/TP_it_*(14)
where *UP_it_* represents the urban population; *TP_it_* represents the total population.

Disaster incidence (*DI*):*DI_it_* = *AD_it_/TAP_it_*(15)
where *AD_it_* represents the agricultural disaster area; *TAP_it_* represents the total agricultural planting area.

The sample period selected in this paper is from 1997 to 2019. Hebei, Shanxi, Inner Mongolia, Liaoning, Jilin, Heilongjiang, Jiangsu, Anhui, Shandong, Henan, Hubei, Guangxi, Sichuan, Guizhou, Yunnan, Shaanxi, Gansu, Ningxia and Xinjiang (19 provinces) are selected as the main corn producing province to calculate their MGP, respectively (Heilongjiang, Jilin, Liaoning and Inner Mongolia are the experimental group, and the other provinces are the control group). In 2019, the maize production of these 19 provinces was 253.31 million tons, accounting for 97.14% of the total maize production in China, and indicating that the selected provinces are well representative. The data sources and descriptive statistics of variables are shown in Table 2. In order to eliminate the impact of inflation, the machinery for maize production, per capita income of rural residents, local agricultural fiscal expenditure, per capita income of urban residents and local fiscal expenditure were reduced by the consumer price index (CPI) based on 1997 to obtain their real values.

## 4. Results and Analysis

### 4.1. Results of MGP and Parallel Trend Test

#### 4.1.1. Results of Maize Green Productivity (MGP)

With the help of the super-epsilon-based measure model (super-EBM) including undesirable output, this study measures the MGP in the main producing areas of maize in China from 1997 to 2019. The efficiency value reflects the sustainability of maize production in the whole and in each province to a certain extent. In terms of overall trend from 1997 to 2019, the average value of MGP in China’s main production provinces first increased, and then decreased before it rebounded slightly. In terms of spatial evolution of each province, the MGP in China’s main production provinces has obvious spatial agglomeration. It gradually forms the distribution pattern centering on the main production areas in the Northeast of China (Heilongjiang, Jilin and Liaoning), Huang-Huai-Hai area (Shandong, Hebei, Henan, Jiangsu and Anhui) and the Southwest of China (Sichuan, Guizhou, Yunnan and Guangxi). The variation range of MGP in the main production areas of Huang-Huai-Hai and Southwest China is significantly higher than that in the Northeast. Especially after 2010, the MGP of the main production areas of Huang-Huai-Hai showed an overall downward trend, while the internal disparities of MGP in Southwest China were significantly intensified. In the main production areas of Northeast China, MGP in Heilongjiang and Jilin province is relatively stable, with a range of less than 5%. The MGP in Liaoning decreased significantly from 1.038 to 0.895 during 1997–2014, and experienced restorative growth after 2015. MGP in Inner Mongolia showed a trend of annual increase, and increased by 38.53% from 1998 to 2019. In the main production areas of Huang-Huai-Hai area, MGP in Jiangsu maintained a high level, with a maximum value of 1.090 (2011), minimum value of 0.989 (2002), and average value of 1.018. MGP in Shandong fluctuated and decreased from 1.011 (1997) to 0.916 (2019). The MGP in Hebei and Henan changed greatly, with a range of more than 30%. In the main production areas in Southwest China, MGP in Sichuan is relatively high, with an average value of 1.005 from 1997 to 2019. The MGP in Guizhou and Guangxi changed greatly, with a range of more than 50%. In other production areas, MGP in Xinjiang increased steadily from 0.910 (1998) to 1.018 (2019). The MGP in Shanxi, Gansu and Shaanxi are relatively low, with average values of 0.812, 0.788 and 0.709, respectively, from 1997 to 2019 (Figure 2).

#### 4.1.2. Parallel Trend Test

Table 3 describes the changes of MGP in the experimental group and control group before and after the implementation of different policies in the two natural experiments. Firstly, in the natural experiment of TPSP, the average value of MGP in the experimental group increased by 2.286% after the implementation of TPSP, and that in the control group increased by 4.661%. The former was significantly lower than the latter, which indicates that the implementation of TPSP may inhibit the improvement of MGP to a certain extent. Secondly, in the natural experiment of PSP, the average value of MGP in the experimental group increased by 1.574% after the implementation of PSP, and that in the control group decreased by 0.633%. The former is significantly higher than the latter, which indicates that the implementation of PSP may promote the improvement of MGP to a certain extent.

This paper draws on the research ideas of Jacobson et al. [70] and Zhong and Peng [71], parallel trend test is required before DID model estimation. The purpose of parallel trend test is to eliminate the interference of the difference between the experimental group and the control group before the implementation of the policy on the DID model. In this paper, the time dummy variables of several years before and after the implementation of the policy are used to replace the difference estimators (DIDTPSP and DIDPSP). By analyzing the dynamic effects of the time dummy variables on MGP, the parallel trend test of DID models of two natural experiments are carried out. The parallel trend test can be expressed:(16)$$MPG_{it}^{TC}=\alpha^{TC}+\sum_{k_{1}\geq -10}^{7}\gamma_{k_{1}}D_{it}^{k_{1}}+\rho^{TC}X_{it}+\nu_{it}^{TC}+\varepsilon_{it}^{TC}$$
(17)$$MPG_{it}^{PS}=\alpha^{PS}+\sum_{k_{2}\geq -6}^{3}\gamma_{k_{2}}D_{it}^{k_{2}}+\rho^{PS}X_{it}+\nu_{it}^{PS}+\varepsilon_{it}^{PS}$$
where $D_{it}^{k_{1}}$ and $D_{it}^{k_{2}}$ represent the time dummy variables for the implementation of TPSP and PSP, respectively. Assuming that the time for the experimental group to implement the TPSP is *year_i_^TPSP^*, let $k_{1}=t-year_{i}^{TPSP}$; for province *i* in the experimental group, when $-10\leq k_{1}\leq 7$, $D_{it}^{k_{1}}=1$, otherwise $D_{it}^{k_{1}}=0$. Similarly, assuming that the time for the experimental group to implement the PSP is *year_i_^PSP^*, let $k_{2}=t-year_{i}^{PSP}$; for province *i* in the experimental group, when $-6\leq k_{2}\leq 3$, $D_{it}^{k_{2}}=1$, otherwise $D_{it}^{k_{2}}=0$.

Figure 3a,b show the results of parallel trend test of TPSP and PSP, respectively. Figure 3a shows that in the natural experiment of TPSP, the coefficient estimates of time dummy variables before 2008 are close to zero, and most of them are not significant; however, they are significantly negative after 2008. This shows that there is no significant difference in MGP between the experimental group and the control group before the implementation of TPSP, but the implementation of this policy has a significant negative impact on MGP. Figure 3b shows that in the natural experiment of PSP, most of the coefficient estimates of time dummy variables before 2016 are not significant, whereas they are significantly positive in 2016 and beyond, indicating that the implementation of PSP has a significant positive impact on MGP.

### 4.2. Natural Experiment of the Temporary Purchase and Storage Policy (TPSP)

#### 4.2.1. Empirical Results and Analysis of the TPSP

In the natural experiment of the TPSP, the paper constructs Model 1, Model 2 and Model 3 to explore the influence of the TPSP on MGP. Among them, Model 1 is the control model without policy variable *DID^TPSP^*, Model 2 is the benchmark model including policy variable *DID^TPSP^*, province-fixed effects and year-fixed effects, and Model 3 is the analysis model including policy variable *DID^TPSP^*, control variables, province-fixed effects and year-fixed effects (as shown in Table 4). In Model 2, the estimated coefficient of the policy variable *DID^TPSP^* is −0.0640, which is significant at the 1% statistical level, reflecting that the implementation of the TPSP had a significant negative impact on MGP. In model 3, after adding control variables such as *GPC*, *EIR*, *AMI*, *UR*, *PA*, *RII*, *DI*, *API* and *AFE*, the estimated coefficient of policy variable *DID^TPSP^* is −0.0321, which is significant at the 5% level, and the Adjusted-R^2^ is significantly higher than model 2; meanwhile, compared with model 1, the addition of *DID^TPSP^* makes the estimated value of each variable coefficient in the model 3 more significant. Therefore, it can be concluded that the implementation of the TPSP had caused the decline of MGP in Heilongjiang, Jilin, Liaoning and Inner Mongolia (experimental group), and weakened the sustainable development capability of maize planting in those provinces.

In order to eliminate the influence of other events on the MGP of the experimental group, the placebo test is used to enhance the stability of the DID analysis. Therefore, it is hypothesized that there are other events that have great negative impacts on the MGP of the experimental group in 2006–2007 or 2009–2010. The paper defines the policy interaction terms *DID^TPSP^* × *year*_2006_, *DID^TPSP^* × *year*_2007_, *DID^TPSP^* × *year*_2009_ and *DID^TPSP^* × *year*_2010_ to indicate that other events occurred in 2006, 2007, 2009 and 2010, which led to the reduction in MGP, thus, Model 4~7 are constructed to perform a placebo test on the DID results (*DID^TPSP^* × *year*_2006_ = 0 in 1997–2005, and *DID^TPSP^* × *year*_2006_ = 1 in 2006–2015; *DID^TPSP^* × *year*_2007_ = 0 in 1997–2006, and *DID^TPSP^* × *year*_2007_ = 1 in 2007–2015; *DID^TPSP^* × *year*_2009_ = 0 in 1997–2008, and *DID^TPSP^* × *year*_2009_ = 1 in 2009–2015; *DID^TPSP^* × *year*_2010_ = 0 in 1997–2009, and *DID^TPSP^* × *year*_2010_ = 1 in 2010–2015), and the placebo test results are shown in Table 4. In Model 4~7, the coefficients of *DID^TPSP^* × *year*_2006_, *DID^TPSP^* × *year*_2007_, *DID^TPSP^* × *year*_2009_ and *DID^TPSP^* × *year*_2010_ are not significant, that is, the decline of MGP in the experimental group is not caused by other events. Therefore, it can be considered that the implementation of the TPSP in 2008 resulted in the loss of MGP in the experimental group.

#### 4.2.2. Effect Analysis of the TPSP

Based on the analysis of influence mechanism, TPSP can affect MGP through support effect, wealth effect, demonstration effect and siphon effect. Combined with the influencing factors of green productivity, this study constructs the interactive terms of *GPC*, *EIR*, *AMI*, *PA*, *RII*, *API*, *AFE* and *DID^TCSP^* to replace the original *DID^TCSP^*. These terms are applied into the model one by one to analyze the impact mechanism of TPSP on MGP. The interaction terms are expressed as *GPC* × *DID^TCSP^*, *EIR* × *DID^TCSP^*, *AMI* × *DID^TCSP^*, *PA* × *DID^TCSP^*, *RII* × *DID^TCSP^*, *API* × *DID^TCSP^* and *AFE* × *DID^TCS^*. The estimation results are shown in Table 5.

In model 8, the estimated coefficient of *GPC* × *DID^TCSP^* is 0.0068 and is significant at the 1% statistical level, which reflects that the increase in production capacity in the experimental group improves MGP under the effect of TPSP. The long-term and stable implementation of TPSP improves farmers’ enthusiasm of production in the experimental group, and sales can be guaranteed under national purchase policy [38]. Therefore, during the implementation of the policy, farmers expanded the scale of production. The increases of marginal productivity brought by large-scale production improve MGP [45], but this improvement effect is relatively small.

In model 9, *EIR* × *DID^TCSP^* has the greatest impact on MGP. The estimated coefficient is 0.1152 and is significant at 5% statistical level, indicating that the improvement of irrigation in the experimental group significantly improved MGP. Because of resource endowment, there is a significant gap in irrigation conditions between the experimental group and control group. In 2015, the average value of *EIR* of the experimental group was 31.11%, while that of the control group was 51.28%. However, the average *EIR* of the experimental group increased by 42.03% from 1997 to 2015, whereas that of the control group increased by only 25.84%. Irrigation condition is an important factor affecting MGP and carbon emission [44,72,73]. Areas in the experimental group lack precipitation, so greater efforts are made to improve of irrigation conditions. Especially at the later stage of TPSP, under the demonstration effect of the policy, these areas led by the Heilongjiang province vigorously developed the water-saving irrigation mode [74]. It greatly improved the irrigation condition in this period (for example, the effective irrigation rate in Heilongjiang increased by 32.17% from 2008 to 2015), and promote the significant improvement of MGP.

In model 10, the estimated coefficient of *AMI* × *DID^TCSP^* is −0.0051, which fails the significance test. It reflects that although the implementation of TPSP has increased machinery input, it cannot significantly improve MGP. This conclusion is consistent with Liao and Huang [37]. The experimental group lags behind in agricultural machinery manufacturing and input (in 2015, the average value of *AMI* in the experimental group was 4.98 KW/HA, while that in the control group was 7.14 KW/HA), and the growth of *AMI* in the experimental group from 1997 to 2015 was slow (the average value of *AMI* in the experimental group increased by 145.64% from 1997 to 2015, and that in the control group increased by 158.65%). Moreover, agricultural machinery in the experimental group has problems such as high energy consumption, outdated machinery, low sowing qualification rate and large harvest damage [75]. These lead to a negative impact of *AMI* on MGP, but it’s not significant. This also reflects that the influence of mechanization level on MGP has a “threshold effect”, and further efforts should be made to develop agricultural machinery manufacturing industry in Northeast China [37].

In model 11, the estimated coefficient of *PA* × *DID^TCSP^* is −0.0107, which fail the significance test. It reflects that the agglomeration effect of maize production formed by TPSP cannot significantly improve MGP. *PA* in the experimental group was significantly higher than that in the control group, but the increase in rice planting area in Northeast China reduced the *PA* of the experimental group year by year. However, the implementation of TPSP slowed down the decline in *PA*. The average *PA* in the experimental group decreased by 19.52% from 1997 to 2007, whereas it decreased by only 6.59% from 2008 to 2015. The slowdown in the decline of *PA* means that the experimental group increased the growing of maize while increasing the growing of rice. Excessive growing of maize eventually led to overcapacity, heavy financial burden, and resources waste [37,38]. Therefore, the *PA* of maize formed by TPSP cannot improve MGP effectively.

In model 12, the coefficient estimate of *RII* × *DID^TCSP^* is −0.1061, which is statistically significant at 5% level. It reflects that although the implementation of TPSP reduces rural income gap, it has a negative impact on MGP. The implementation of this policy has guaranteed farmers’ income and has obvious wealth effect [37], thus encouraging farmers to obtain more output by expanding production scale and increasing production material input. With continuous rise of purchase price, farmers’ income is directly related to output. In their pursuit of high income, many problems such as excessive use of production materials, overcapacity, and overdraft of natural environment occurred [37], which lead to a significant decrease in MGP.

In model 13, the coefficient estimate of *API* × *DID^TCSP^* is −0.0323, which is statistically significant at 5% level. It reflects the increase in *API* caused by the implementation of TPSP hinders the improvement of MGP in the experimental group. Under the support effect and wealth effect, the overuse of production materials caused by the implementation of TPSP drives the rise of *API*, through the mediation of supply and demand effect of the raw material market [37,76]. The rise of *API* increases the cost of food production, and to a certain extent affects farmers’ use of machinery leasing, investment in high-quality seeds, and application of energy-saving and emission-reducing technologies. Therefore, the rise of *API* hinders the improvement of MGP in the experimental group.

In model 14, the estimated coefficient of *AFE* × *DID^TCSP^* is 0.0398, which is statistically significant at 5% level. It reflects the increase in *AFE* driven by the implementation of TPSP contributes to the improvement of MGP to a certain extent. TPSP has a strong guidance effect [55,77]. The implementation of the policy in the experimental group increases the financial support of the local government for agricultural production, and increases the proportion of agricultural finance. The average of *AFE* of the experimental group increased from 7.80% to 8.73% (an increase of 11.97%) during 1997–2007, and that of the control group increased from 8.66% to 9.60% (an increase of 10.93%). The average *AFE* of the experimental group increased from 9.17% to 13.87% (an increase of 51.26%) during 2008–2015, and that of the control group increased from 10.21% to 12.67% (an increase of 24.02%). The increase in *AFE*, especially the increase in subsidies to support grain production, such as direct grain subsidies, comprehensive subsidies for agricultural materials, improved seed subsidies and agricultural machinery purchase subsidies, increase the income of maize production, utilization rate of improved species and input of agricultural machinery, thereby improving MGP.

### 4.3. Natural Experiment of the Producer Subsidy Policy (PSP)

#### 4.3.1. Empirical Results and Analysis of the PSP

In the natural experiment of the PSP, the paper also constructs the control model (Model 15), benchmark model (Model 16) and analysis model (Model 17) to explore the influence of the PSP on MGP. In Model 16, the estimated coefficient of the policy variable *DID^PSP^* is 0.0735, which is significant at the 1% level, reflecting that the implementation of the PSP has a significant positive impact on MGP. In model 17, after adding control variables, the estimated coefficient of policy variable *DID^PSP^* becomes 0.0807 (1% level), and the Adjusted-R^2^ is significantly higher than model 16; meanwhile, compared with model 15, the addition of *DID^PSP^* makes the estimation results more robust. Therefore, it can be concluded that the implementation of the PSP has improved the MGP of the experimental group, and enhanced the sustainable development capability of maize planting in those provinces.

In the placebo test, it is hypothesized that other events significantly increased MGP of the experimental group in 2014–2015 or 2017–2018. The paper defines the policy interaction terms *DID^PSP^* × *year*_2014_, *DID^PSP^* × *year*_2015_, *DID^PSP^* × *year*_2017_ and *DID^PSP^* × *year*_2018_ to indicate that other events occurred in 2014, 2015, 2017 and 2018, which led to the improvement of MGP, thus, Model 18~21 are constructed to perform a placebo test on the DID results (*DID^PSP^* × *year*_2014_ = 0 in 2008–2013, and *DID^PSP^* × *year*_2014_ = 1 in 2014–2019; *DID^PSP^* × *year*_2015_ = 0 in 2008–2014, and *DID^PSP^* × *year*_2015_ = 1 in 2015–2019; *DID^PSP^* × *year*_2017_ = 0 in 2008–2016, and *DID^PSP^* × *year*_2017_ = 1 in 2017–2019; *DID^PSP^* × *year*_2018_ = 0 in 2008–2017, and *DID^PSP^* × *year*_2018_ = 1 in 2018-2019), and the placebo test results are shown in Table 6. In Model 18~21, the coefficients of *DID^PSP^* × *year*_2014_, *DID^PSP^* × *year*_2015_, *DID^PSP^* × *year*_2017_ and *DID^PSP^* × *year*_2018_ are not significant, that is, the improvement of MGP is not caused by other events. Therefore, it can be considered that the implementation of the PSP in 2016 lead to the improvement of MGP in the experimental group.

#### 4.3.2. Parallel Trend Test

The interaction terms *GPC* × *DID^PSP^*, *EIR* × *DID^PSP^*, *AMI* × *DID^PSP^*, *PA* × *DID^PSP^*, *RII* × *DID^PSP^*, *API* × *DID^PSP^*, *AFE* × *DID^PSP^* are constructed, respectively, to replace the original *DID^PSP^*. They are then applied into the models to obtain estimation results, which can be used to analyze the impact mechanism of PSP on MGP. The estimated results are shown in Table 7.

In model 22, the estimated coefficient of *GPC* × *DID^PSP^* is −0.0017, which fail the significance test. It shows that the increase in *GPC* may hinder the improvement of MGP during the implementation of PSP. The purpose of producer subsidy reform is to reduce excess capacity according to market demand and release inventory pressure [35,39]. However, the problem of overcapacity in the experimental group is still serious now. Appropriate adjustment of grain production structure and reduction in production capacity is the key to the improvement of green productivity.

In model 23, the estimated coefficient of *EIR* × *DID^PSP^* is 0.1008, which is significant at 1% statistical level. It reflects that the improvement of irrigation conditions in the experimental group is still an important factor to improve MGP in this period. Under the demonstration effect of the PSP, the gap between the experimental group and the control group in irrigation conditions continues to narrow. The average value of *EIR* in the experimental group increased by 3.51% during 2016–2019, and that control group increased by 2.55%. Therefore, the continuous improvement of irrigation conditions in the experimental group promotes the improvement of MGP.

In model 24, the estimated coefficient of *AMI* × *DID^PSP^* is −0.0073, which is significant at 5% statistical level. It shows that the improvement of agricultural machinery level in this period cannot improve MGP. The average value of *AMI* in the experimental group increased by 29.72% from 2008 to 2019, despite this remarkable increase, the experimental group still lagged behind the control group in agricultural machinery manufacturing and investment (in 2019, the average value of *AMI* in the experimental group was only 5.05 KW/HA, whereas that of the control group was 6.13 KW/HA). Moreover, the problems of out-dated agricultural machinery and low work efficiency in the experimental group still exist, so *AMI* has a negative impact on MGP during this period.

In model 25, the estimated coefficient of *PA* × *DID^PSP^* is 0.0287, which is significant at 1% statistical level. It reflects that *PA* caused by the support effect and siphon effect of PSP improves MGP. The PSP is a policy-based purchase and storage reform that releases the vitality of the market. The subsidy is based on market pricing system, which is decoupled from maize output and strengthens the market regulation of maize production [39,77]. Therefore, producers “supported” and “attracted” by the support effect and siphon effect of PSP have a higher level in production technology, resource management and environmental protection awareness, which promotes the improvement of MGP.

In model 26, the estimated coefficient of *RII* × *DID^PSP^* is 0.0601, which is significant at 1% statistical level. It reflects the increase in farmers’ income caused by the wealth effect of PSP can promote MGP. The implementation of PSP decouples subsidy income from grain output, and subsidy behavior does not directly affect grain output and price [39,77]. When PSP increases farmers’ income, it can alleviate the problems of overcapacity, excessive use of production means, and overdraft of natural environment. Moreover, with the increase in income, farmers pay more attention to the learning of scientific knowledge, the application of advanced technology and improvement of management ability [15,78]. Therefore, after the implementation of PSP, the increase in farmers’ income improved MGP.

In model 27, the estimated coefficient of *API* × *DID^PSP^* is 0.0335, which is significant at the 5% statistical level. It reflects the rise in *API* caused by the PSP promotes the improvement of MGP in the experimental group. With the support effect and wealth effect of PSP, producers have gradually changed their original concept of pursuing output [77]. With the rising cost of grain production, they pay more attention to reasonable arrangement of production plan, adjustment of planting structure and improvement of resource utilization efficiency. Therefore, the rise of *API* in the experimental group improves the MGP.

In model 28, the estimated coefficient of *AFE* × *DID^PSP^* is 0.2861, which is significant at 1% statistical level. It reflects that the increase in *AFE* driven by the demonstration effect of PSP can greatly improve MGP. Since the implementation of PSP in the experimental group, the proportion of agricultural financial expenditure has been relatively stable. The average *AFE* in 2019 was 14.46%, an increase of 4.24% compared with that in 2015. Moreover, abolishing TPSP and implementing PSP is the pioneer of structural reform of grain supply side, which has significant demonstration effect and guiding significance [77]. The local government (in the experimental group) actively promote the structural reform of grain supply side, realized the market-based procurement of maize, and provided better policy environment for improving MGP with agricultural science and technology incubation policy, rural financial support policy, agricultural support and protection subsidies.

## 5. Discussions

Due to the different historical background, objectives and measures of TPSP and PSP (as shown in Table 1), the influence of the two policies on MGP is also significantly different. The focus of TPSP is to stabilize the enthusiasm of grain production and improve grain output. The PSP, however, focuses on solving the problems of maize overcapacity, high inventory pressure, heavy financial burden, waste of resources and environmental pollution, on the premise of secure grain supply, so as to improve productivity, optimize resource allocation, adjust food structure and strengthen sustainable production. The following conclusion can be draw from the comparison of the different impact on MGP of each effect during the period of TPSP and PSP.

Under the support effect of TPSP, the increase in grain production capacity has improved MGP to a certain extent, but under the support effect of PSP, the increase in grain production capacity has a negative impact on MGP (although the estimated coefficient is not significant). The different impact of grain production capacity reflects that continued expansion of production capacity will hinder the improvement of MGP. Therefore, controlling the production capacity within a reasonable range is necessary to promote the development of green production of maize, which verifies the conclusion of Xu et al. [45] and Coderoni and Vanino [79].

Due to the strong guiding characteristics of grain policy [54,55], both TPSP and PSP show significant demonstration effects, and affect MGP by improving the agricultural production condition and increasing agricultural financial expenditure. In terms of agricultural production conditions, the improvement of irrigation system has always been an important factor to improve MGP [44,72], but the lag in the development of agricultural machinery in the experimental group hinders the improvement of MGP, which is consistent with Liao and Huang [37] and He et al. [80]. Therefore, improving the productivity and working efficiency of agricultural machinery in the experimental group is expected to further improve MGP. In terms of agricultural financial expenditure, PSP is the pioneer of structural reform on the food supply side, and under its demonstration effect, agricultural financial expenditure of the experimental group also tends to support the efficient utilization of resources and development of green agriculture [51]. Therefore, the positive impact of agricultural financial expenditure on MGP is significantly improved.

Both TPSP and PSP can effectively improve farmers’ income, and both policies show significant wealth effect. Under the TPSP, farmers’ pursuit of high income has induced such problems as overcapacity, excessive use of production means and damage of natural environment [39,77], which significantly reducing MGP. Moreover, the rise in the price of production means caused by excessive use has affected farmers’ investment in machinery leasing, improved seeds and the application of energy-saving and emission-reducing technologies [15], and further hinders the improvement of MGP. The PSP does not directly affect grain output and price, which alleviates the problems of excessive use of production means and damage to the natural environment [39,77]. Moreover, the policy guided signal released by this reform has further promoted the energy-saving and emission-reducing technology [51]. In addition, the negative feedback mediation of the price of production means has prompted farmers to change their production concept and pay more attention to the rational arrangement of production, adjustment of the planting structure and improvement of resource utilization efficiency.

Production agglomeration has a negative impact on MGP under the siphon effect of TPSP, whereas it has a positive impact on MGP under the siphon effect of PSP. This difference is caused by the different producers and investors it attracts: The capital, technology, labor and other production elements in the surrounding areas attracted by TPSP are concentrated in improving maize production [46], whereas the producers and investors attracted by the PSP have competitive advantages in production technology, resource management and environmental protection awareness (since producer subsidy is far lower than TPSP in increasing income, inefficient producers will be unable to obtain more benefits by expanding production), which is consistent with Yang et al. [80]. The improvement of MGP by the production subject jointly promotes the improvement of overall MGP in the experimental group.

## 6. Conclusions and Implications

### 6.1. Conclusions

This study expounds the evolutionary logic of maize purchase and storage policy, and analyzes the factors influencing grain green productivity and the effect of grain purchase and storage policy; then, the super-epsilon-based measure model (super-EBM) is adopted to measure maize green productivity in each province based on the data of production elements in main producing areas from 1997 to 2019; finally, two groups of DID models are constructed to study the influence of temporary purchase and storage policy and producer subsidy policy on maize green productivity. The main conclusions include:(1)The implementation of temporary purchase and storage policy reduces maize green productivity in the experimental group (Heilongjiang, Jilin, Liaoning and Inner Mongolia), whereas the implementation of producer subsidy policy improves maize green productivity in the experimental group, this is due to the difference of policy effects under the different regulatory objectives and measures.(2)Under the demonstration effect of temporary purchase and storage policy or producer subsidy policy, the improvement of effective irrigation and the increase in agricultural financial expenditure are important factors to improve maize green productivity. The lag of the experimental group in the development of agricultural mechanization has been hindering the development of maize green production.(3)After the shift from temporary purchase and storage policy to producer subsidy policy, continuous increase in production capacity hinders the improvement of maize green productivity under the support effect of two policies. Under the wealth effect of two policies, the influence of farmers’ income and agricultural production price on the maize green productivity shifts from negative to positive because of the change of farmers’ production concept and technical improvement. Under the siphon effect of two policies, the influence of production agglomeration on maize green productivity shifts from negative to positive because of the competitive advantages of the attracted producers in production technology, resource management and environmental protection consciousness.

### 6.2. Policy Implications

According to the content of the discussion and conclusion, the policy implications mainly include:

Firstly, maintaining moderate scale management, and adjusting the grain structure. The government can improve the socialized service system, accelerate the cultivation of new agricultural business entities, and maintain moderate scale management of agricultural production; the government ought to promote the green and technological process of agricultural production, and build a high-quality, green and sustainable agricultural system through policy and technological innovation, to strengthen scientific and technological support, thus promoting cost saving and efficiency increase in grain production; grain production should be oriented by market demand, in order to promote the matching of grain production structure with market demand, thus realizing the transformation of grain production to high-quality and sustainable.

Secondly, improving the agricultural irrigation level, and enhancing the agricultural mechanization degree. The government should strengthen the research on farmland water conservancy irrigation technology, and reasonably introduce advanced irrigation technologies such as drip irrigation, micro-sprinkler irrigation, and the integration of water, fertilizer and pesticide; the government should increase investment in irrigation facilities construction, meanwhile cooperate with water price system reform, compensation system and other measures to protect farmland construction and upgrading of water conservancy irrigation facilities. In addition, the government should accelerate the upgrading process of agricultural machinery; meanwhile, increase the level of subsidies and personnel training for agricultural mechanization, in order to improve the efficiency and quality of agricultural machinery and equipment.

Finally, improving the agricultural subsidy mechanism, and increasing the intensity of production subsidies. The subsidy system for agricultural producers and related legal systems should be further improved, meanwhile adjust and improve the agricultural policy system according to the actual level of China’s agricultural development and the WTO agricultural agreement; the completing of subsidy mechanism should primarily adhere to the orientations of “Green Box Policy”, and then strengthen subsidies and adjust the subsidy structure. The government should standardize the management mechanism of agricultural production subsidies, give full play to the roles of financial, auditing and supervisory organs, and urge all departments to implement the distribution of funds in an open, transparent and standardized manner.

## Figures and Tables

**Figure 1 ijerph-19-06310-f001:**
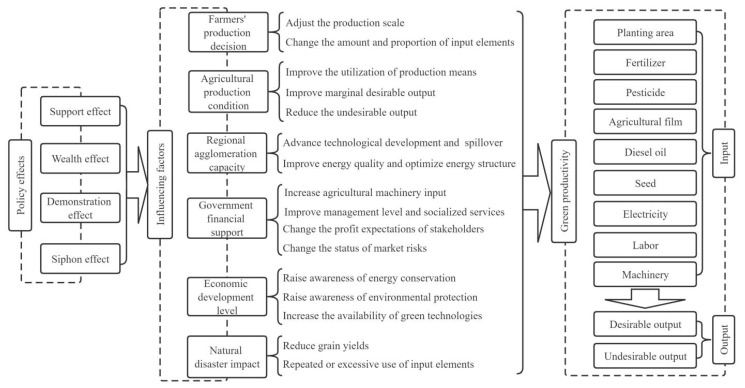
Influence mechanism and policy effects.

**Figure 2 ijerph-19-06310-f002:**
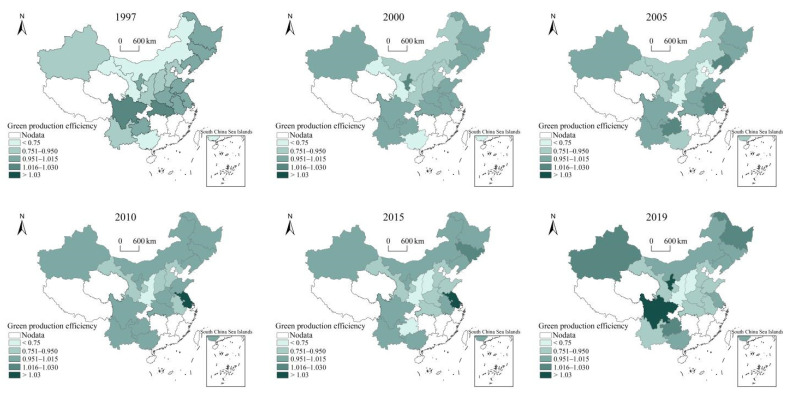
Spatial evolution of MGP from 1997 to 2019.

**Figure 3 ijerph-19-06310-f003:**
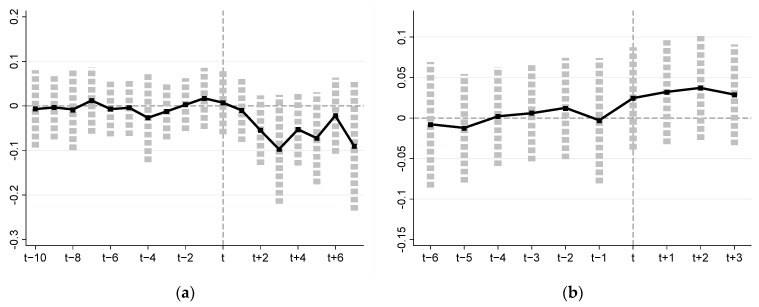
Spatial evolution of MGP from 1997 to 2019. (**a**) Parallel trend test before and after TPSP implementation; (**b**) parallel trend test before and after PSP implementation Note: The abscissa represents the time dummy variables, and the ordinate represents the coefficient estimates of the time dummy variables.

**Table 1 ijerph-19-06310-t001:** Maize purchase and storage policies implemented in China from 1997 to 2019.

Policy	Background	Objectives	Measures	Impacts
Protective price policy (PPP) 1997–2007	With monopolistic purchase by the government, the purchase price is higher than sales price. The purchased grain cannot achieve profitable sales. Therefore, the local governments have heavy financial burden, and state-owned grain enterprises face serious loss [34].	The policy aims to adjust grain production structure on the premise of maintaining farmers’ enthusiasm for grain production. It is also expected to achieve profitable sales of policy grain reserves and reduce the financial burden of the government.	Protective price is implemented for the purchasing price, and the local governments make purchase without limitation. The areas implementing PPP is reduced and the level of protective price is lowered.	In the initial stage of the implementation of the policy, the enthusiasm for grain production and sales was improved, and the loss situation of state-owned enterprises took a turn for the better [35]. However, in the later stage, monopolistic purchase slowed down the growth of farmers’ income and decreased their enthusiasm for production, which led to the instability of investment in grain production and the low productivity [35,36].
Temporary purchase and storage policy (TPSP) 2008–2015	In the later stage of its implementation, PPP led to unstable grain output, low productivity and slowdown of the growth rate of farmers’ income [35,36]. Additionally, the mismatch between domestic grain production and consumption, together with the sharp decline of international grain prices, led to the low price of maize in the domestic market and the lack of farmers’ enthusiasm for production [35,37].	It aims to improve overall productivity, increase the purchase price to stabilize the farmers’ enthusiasm for grain production, strengthen government regulation of the grain market to guarantee the security of grain supply.	TPSP is implemented in Heilongjiang, Jilin, Liaoning and Inner Mongolia, with an increase in purchase price each year. Moreover, a competitive auction system is formulated to regulate the sales of policy grain.	Grain prices regulation system was improved. Grain output increased steadily, and farmers’ income grew rapidly [37]. However, the continuous implementation of TPSP and the annual increase in purchase prices also led to such problems as overcapacity of maize, increased pressure for inventory and import, heavy financial burden, frequent factor mismatch and resource waste [37,38].
Producer subsidy policy (PSP) 2016–2019	As international grain prices continued to decline and domestic grain production costs increased, implementation of TPSP led to higher domestic price of maize than the international price, and the import volume of maize remains high [35,36]. Moreover, overcapacity of maize stimulated by the policy led to continuous increase in temporary storage inventory and financial pressure [35,39].	The policy aims to alleviate the problems of maize overcapacity, increased inventory and import pressure and heavy financial burden. It also expects to reduce factor mismatch and resource waste, and optimize the grain production structure to match supply with demand.	TPSP for maize in Heilongjiang, Jilin, Liaoning and Inner Mongolia is abolished. The support policy of “market pricing + producer subsidy” is implemented.	Initial effect was achieved in adjustment of grain production structure. Maize production capacity was controlled. The inventory pressure was effectively released. Import volume decreased significantly. Productivity was improved, and the problems of factor mismatch and resource waste were alleviated [39].
Summary	The government monopoly purchase, protective price purchase and temporary purchase can directly lead to a substantial increase in the grain purchase price, which makes it difficult for the purchased grain to achieve profitable sales, and results in overcapacity, resource waste, inventory pressure, financial burden and the lack of international competitiveness. However, these reasons have jointly promoted the continuous adjustment and reform of China’s grain policy.	The goal of the Chinese government to adjust the grain purchase and storage policies is to realize the market-oriented reform on the basis of ensuring the grain supply security and farmers’ enthusiasm for production, further release market vitality, improve production efficiency, reduce resource waste and carbon emissions, and finally improve the sustainability of grain production.	In different periods, the Chinese government has adopted different purchase and storage measures to ensure the realization of policy objectives. From the various purchase and storage measures included in PPP, TPSP and PSP, the implementation and adjustment of different measures always focus on reducing government intervention and strengthening market forces; and the continuous adjustment and upgrading of various measures are to achieve sustainable security of food supply.	In the early stage of PPP, TPSP and PSP, significant effects have been achieved in stabilizing grain planting income and farmers’ production enthusiasm; however, due to the strong policy intervention of PPP and TPSP, problems such as resource waste, excessive financial burden and low production efficiency occurred in the later stage of PPP and TPSP. As the pioneer in the structural reform of the grain supply side, the implementation of PSP has effectively reduced excess production capacity, improved the problem of resource waste, and is more in line with the concept of sustainable development of the grain industry.

**Table 2 ijerph-19-06310-t002:** Data sources and descriptive statistics of variables.

Variable	Units	Mean	Maximum	Minimum	Std. Dev.	Data Source
Maize production	10^4^ t	922.302	4280.190	82.000	826.224	Data from the National Bureau of Statistics of China
Maize planting area	10^4^ ha	165.883	736.115	13.110	130.298	
Staple grain production	10^4^ t	2134.275	7022.560	218.900	1456.767	
Total power of agricultural machinery	10^4^ kW	3420.046	13,353.020	288.400	2827.475	
Total agricultural planting area	10^4^ ha	683.271	1490.272	97.760	327.212	
Total effective irrigated area	10^4^ ha	254.983	617.759	34.359	147.818	
Total cultivated area	10^4^ ha	594.473	1586.410	110.706	247.190	
Temperature fluctuation index	—	0.011	0.293	−0.235	0.068	
Precipitation fluctuation index	—	−0.002	2.736	−0.613	0.267	
Sunshine change index	—	−0.001	0.256	−0.374	0.090	
Agricultural production price index	—	1.030	1.281	0.919	0.059	
Fertilizer for maize production	10^4^ t	54.190	228.033	5.208	44.071	Data from the Compilation of Cost-Benefit Data of Agricultural Products in China (1998–2020)
Pesticide for maize production	10^4^ t	0.030	0.150	0.002	0.033	
Agricultural film for maize production	10^4^ t	0.909	7.924	0.006	1.384	
Diesel oil for maize production	10^4^ t	0.320	4.247	0.001	0.503	
Seed for maize production	10^4^ t	6.303	25.551	0.499	4.746	
Electricity for maize production	10^4^ kWh	78,591.310	576,021.100	65.640	100,492.700	
Labor for maize production	10^4^ day	19,480.860	59,891.500	2163.150	11,048.620	
Labor for staple grain production	10^4^ day	47,034.500	189,096.800	4079.624	28,468.230	
Machinery for maize production	10^4^ CNY	175,169.900	1,562,619.000	59.084	265,001.900	
Per capita income of rural residents	10^4^ CNY	0.585	2.268	0.119	0.432	Data from the China Rural Statistical Yearbook (1998–2020)
Agricultural disaster area	10^4^ ha	164.861	739.370	2.910	113.166	
Local agricultural fiscal expenditure	10^8^ CNY	284.575	1310.890	3.925	300.258	
Per capita income of urban residents	10^4^ CNY	1.603	5.106	0.359	1.045	Data from the Statistical Yearbooks of each province
Local fiscal expenditure	10^8^ CNY	2458.199	12,573.550	33.630	2502.075	
Urban population	10^4^ people	2045.297	6194.000	168.718	1252.349	
Total population	10^4^ people	4862.998	10,106.000	530.000	2506.600	

Note: “—” represent no data.

**Table 3 ijerph-19-06310-t003:** Changes in the mean value of MGP before and after the implementation of TPSP and PSP.

Variable Name	Policies	Groups	Periods			Differences
			1997–2007	2008–2015	2016–2019	
Mean value of MGP	TPSP	Experimental group 1	0.9753	0.9976	—	0.0223
		Control group 1	0.9204	0.9633	—	0.0429
		Difference 1	0.0549	0.0343	—	−0.0206
	PSP	Experimental group 2	—	0.9976	1.0133	0.0157
		Control group 2	—	0.9633	0.9572	−0.0061
		Difference 2	—	0.0343	0.0561	0.0218

Note: “—” represent no data.

**Table 4 ijerph-19-06310-t004:** DID analysis results of the TPSP.

Variables	Model 1	Model 2	Model 3	Model 4	Model 5	Model 6	Model 7
*DID^TPSP^* × *year*_2008_	—	−0.0640 *** (0.0079)	−0.0321 ** (0.0160)	—	—	—	—
*DID^TPSP^* × *year*_2006_	—	—	—	−0.0208 (0.0164)	—	—	—
*DID^TPSP^* × *year*_2007_	—	—	—	—	−0.0175 (0.0158)	—	—
*DID^TPSP^* × *year*_2009_	—	—	—	—	—	−0.0291 (0.0217)	—
*DID^TPSP^* × *year*_2010_	—	—	—	—	—	—	−0.0316 (0.0271)
*GPC*	0.0047 (0.0039)	—	0.0046 * (0.0027)	0.0041 (0.0028)	0.0039 (0.0027)	0.0044 (0.0027)	0.0049 * (0.0026)
*EIR*	0.1260 *** (0.0274)	—	0.1524 *** (0.0282)	0.1544 *** (0.0281)	0.1546 *** (0.0281)	0.1533 *** (0.0281)	0.1492 *** (0.0282)
*AMI*	−0.0133 *** (0.0031)	—	−0.0099 *** (0.0021)	−0.0099 *** (0.0022)	−0.0101 *** (0.0022)	−0.0099 *** (0.0022)	−0.0099 *** (0.0021)
*UR*	0.0656 (0.0433)	—	0.0573 (0.0467)	0.0573 (0.0469)	0.0559 (0.0468)	0.0573 (0.0468)	0.0581 (0.0467)
*PA*	0.0063 (0.0085)	—	−0.0111 (0.0091)	−0.0112 (0.0091)	−0.0107 (0.0092)	−0.0111 (0.0092)	−0.0107 (0.0091)
*RII*	0.3275 *** (0.1003)	—	−0.1369 * (0.0819)	−0.1330 (0.0888)	−0.1413 * (0.0870)	−0.1369 (0.0878)	−0.1420 * (0.0840)
*DI*	−0.0774 * (0.0394)	—	−0.1076 *** (0.0391)	−0.1044 *** (0.0390)	−0.1042 *** (0.0391)	−0.1049 *** (0.0390)	−0.1126 *** (0.0396)
*TF*	−0.1434 (0.1654)	—	−0.1641 (0.1679)	−0.1486 (0.1660)	−0.1499 (0.1663)	−0.1696 (0.1708)	−0.1731 (0.1693)
*PF*	−0.0258 (0.0194)	—	−0.0248 (0.0192)	−0.0258 (0.0191)	−0.0255 (0.0192)	−0.0255 (0.0191)	−0.0249 (0.0190)
*SC*	−0.0478 (0.0662)	—	−0.0336 (0.0646)	−0.0359 (0.0645)	−0.0354 (0.0646)	−0.0338 (0.0645)	−0.0330 (0.0644)
*API*	0.0988 (0.0771)	—	−0.0627 * (0.0366)	−0.0607 (0.0448)	−0.0638 * (0.0378)	−0.0458 (0.0498)	−0.0508 (0.0486)
*AFE*	0.2788 (0.3098)	—	0.0274 ** (0.0136)	0.0127 (0.0239)	0.0244 (0.0185)	0.0325 * (0.0182)	0.0306 * (0.0177)
Province-fixed effects	Control	Control	Control	Control	Control	Control	Control
Year-fixed effects	Control	Control	Control	Control	Control	Control	Control
*Adj*-*R*^2^	0.543	0.286	0.698	0.648	0.647	0.649	0.651

Note: the standard error of coefficient estimation is shown in brackets, ‘*’, ‘**’, ‘***’ represent the significance levels of 10%, 5% and 1%, respectively; “—” represent no data.

**Table 5 ijerph-19-06310-t005:** Mechanism analysis results of the TPSP.

Variables	Model 8	Model 9	Model 10	Model 11	Model 12	Model 13	Model 14
*GPC* × *DID^TPSP^*	0.0068 *** (0.0018)	—	—	—	—	—	—
*EIR* × *DID^TPSP^*	—	0.1152 ** (0.0521)	—	—	—	—	—
*AMI* × *DID^TPSP^*	—	—	−0.0051 (0.0032)	—	—	—	—
*PA* × *DID^TPSP^*	—	—	—	−0.0107 (0.0076)	—	—	—
*RII* × *DID^TPSP^*	—	—	—	—	−0.1061 ** (0.0447)	—	—
*API* × *DID^TPSP^*	—	—	—	—		−0.0323 ** (0.0154)	—
*AFE* × *DID^TPSP^*	—	—	—	—	—	—	0.0398 ** (0.0155)
Control variables	Control	Control	Control	Control	Control	Control	Control
Province-fixed effects	Control	Control	Control	Control	Control	Control	Control
Year-fixed effects	Control	Control	Control	Control	Control	Control	Control
*Adj*-*R*^2^	0.660	0.613	0.627	0.646	0.647	0.651	0.653

Note: the standard error of coefficient estimation is shown in brackets, ‘**’, ‘***’ represent the significance levels of 5% and 1%, respectively; “—” represent no data.

**Table 6 ijerph-19-06310-t006:** DID analysis results of the PSP.

Variables	Model 15	Model 16	Model 17	Model 18	Model 19	Model 20	Model 21
*DID^PSP^*	—	0.0735 *** (0.0073)	0.0807 *** (0.0222)	—	—	—	—
*DID^PSP^* × *year*_2014_	—	—	—	0.0652 (0.0423)	—	—	—
*DID^PSP^* × *year*_2015_	—	—	—	—	0.0661 (0.0442)	—	—
*DID^PSP^* × *year*_2017_	—	—	—	—	—	0.0650 (0.0465)	—
*DID^PSP^* × *year*_2018_	—	—	—	—	—	—	0.0652 (0.0504)
*GPC*	−0.0046 (0.0032)	—	−0.0059 ** (0.0024)	−0.0090 *** (0.0032)	−0.0075 *** (0.0028)	−0.0063 ** (0.0028)	−0.0057 * (0.0029)
*EIR*	0.1231 *** (0.0278)	—	0.1243 *** (0.0241)	0.1428 *** (0.0281)	0.1330 *** (0.0275)	0.1261 *** (0.0276)	0.1247 *** (0.0276)
*AMI*	−0.0072 ** (0.0034)	—	−0.0064 *** (0.0021)	−0.0079 ** (0.0032)	−0.0075 ** (0.0029)	−0.0074 ** (0.0030)	−0.0075 ** (0.0031)
*UR*	0.2315 * (0.1282)	—	0.1747 * (0.1039)	0.1135 (0.1271)	0.1485 (0.1216)	0.1930 (0.1233)	0.2080* (0.1248)
*PA*	0.0339 *** (0.0088)	—	0.0312 *** (0.0086)	0.0292 *** (0.0083)	0.0307 *** (0.0085)	0.0323 *** (0.0088)	0.0326 *** (0.0089)
*RII*	0.0892 *** (0.0351)	—	0.1164 *** (0.0354)	0.1208 *** (0.0365)	0.1122 *** (0.0346)	0.1037 *** (0.0341)	0.0989 *** (0.0340)
*DI*	−0.1093 ** (0.0487)	—	−0.1249 *** (0.0466)	−0.1291 ** (0.0498)	−0.1274 *** (0.0480)	−0.1239 ** (0.0475)	−0.1207 ** (0.0472)
*TF*	0.0806 (0.0708)	—	−0.0514 (0.0702)	0.0007 (0.0741)	0.0240 (0.0726)	0.0501 (0.0707)	0.0613 (0.0710)
*PF*	0.0103 (0.0235)	—	−0.0032 (0.0233)	0.0158 (0.0232)	0.0079 (0.0230)	0.0099 (0.0232)	0.0065 (0.0233)
*SC*	−0.0826 (0.0688)	—	−0.0926 (0.0680)	−0.0833 (0.0666)	−0.0850 (0.0660)	−0.0859 (0.0664)	−0.0816 (0.0667)
*API*	0.0175 (0.0166)	—	0.0106 (0.0082)	0.0490 (0.0368)	0.0143 (0.0870)	−0.0088 (0.0080)	−0.0089 (0.0184)
*AFE*	0.3393 *** (0.1117)	—	0.2714 *** (0.0588)	0.462 *** (0.1338)	0.3127 *** (0.1016)	0.3462 *** (0.1087)	0.3637 *** (0.1138)
Province-fixed effects	Control	Control	Control	Control	Control	Control	Control
Year-fixed effects	Control	Control	Control	Control	Control	Control	Control
*Adj*-*R*^2^	0.757	0.370	0.784	0.729	0.731	0.739	0.736

Note: the standard error of coefficient estimation is shown in brackets, ‘*’, ‘**’, ‘***’ represent the significance levels of 10%, 5% and 1%, respectively; “—” represent no data.

**Table 7 ijerph-19-06310-t007:** Mechanism analysis results of the PSP.

Variables	Model 22	Model 23	Model 24	Model 25	Model 26	Model 27	Model 28
*GPC* × *DID^PSP^*	−0.0017 (0.0015)	—	—	—	—	—	—
*EIR* × *DID^PSP^*	—	0.1008 *** (0.0342)	—	—	—	—	—
*AMI* × *DID^PSP^*	—	—	−0.0073 ** (0.0033)	—	—	—	—
*PA* × *DID^PSP^*	—	—	—	0.0287 *** (0.0086)	—	—	—
*RII* × *DID^PSP^*	—	—	—	—	0.0601 *** (0.0246)	—	—
*API* × *DID^PSP^*	—	—	—	—		0.0335 ** (0.0162)	—
*AFE* × *DID^PSP^*	—	—	—	—	—	—	0.2861 *** (0.1009)
Control variables	Control	Control	Control	Control	Control	Control	Control
Province-fixed effects	Control	Control	Control	Control	Control	Control	Control
Year-fixed effects	Control	Control	Control	Control	Control	Control	Control
*Adj*-*R*^2^	0.819	0.782	0.824	0.733	0.773	0.769	0.837

Note: the standard error of coefficient estimation is shown in brackets, ‘**’, ‘***’ represent the significance levels of 5% and 1%, respectively; “—” represent no data.

## Data Availability

The datasets used and analyzed during the current study are available from the corresponding author on reasonable request.

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
