# Peer review of "Can the Adjustment of China’s Grain Purchase and Storage Policy Improve Its Green Productivity?"

_ijerph, 2022, doi:10.3390/ijerph19106310_

Round 1
Reviewer 1 Report
I want to congratulate the authors of the article for the extensive and exhaustive research they have carried out. Also, thank them for the explanation of the different formulas used since with them it has been easier to understand their research.
Here are some comments for you to consider:
-Line 15: explain the acronym EBM.
-Line 16: explain the acronym DID.
-Reorganize the abstract: emphasize the objective of the article, comment on the areas of study (do not put it in the conclusions part of the abstract), comment on the results obtained from the research. Lines 12 and 13 are fine as an introduction and lines 28 to 30 are fine as a conclusion, but I would rearrange/rewrite lines 14 to 27.
-Line 31 (keywords): sort alphabetically.
-Line 91: change the14th to the 14th.
-Line 150 (Table 1): summarize each column, be more specific, put outline when possible.
-Line 160: change [43-44] to [43,44] since there are only two references.
-Line 178: same as 160.
-Lines 300 to 301 ((9)): a vertical line appears, delete
-Line 401 (figure 3): put the parameters of the X-axis in superscript: t-6 --> t-6 ; t+1 --> t+1
-Line 495: same as line 160 and 178.
-Line 500 to 505: remove reference 37 in line 500 as it reappears and reference it in line 504.
-Line 757 (references): check references number 17 (put all the authors) and 19 (check the way to put the authors),
Thanks,
Author Response
We would like express our deep gratitude for your comments. We made point-to-point revisions according to the review comments. Please check the attached file for details.
Thank you very much for your consideration.
Best regards!

Reviewer 2 Report
The study aims to evaluate the effects of the temporary purchase and storage policy (TPSP) and the producer subsidy policy (PSP) on Chinese maize green productivity (MGP) between 1997 and 2019. While MGP is measured using the Super-EBM model, the effects of TSSP and PSP are evaluated through the construction of two groups of DID. The results of this approach are discussed and lead to several findings and some policy suggestions.
At first glance the article is well built and interesting. The-state-of-the-art is well described in Introduction with even very recent citations, while background comes clear in the second part “Analysis of policy evolution and influence mechanism”, “results and analysis” are well discussed, maybe too long discussed, while “conclusions and implications” are interesting.
Part 3. “Materials and Methods” should be revised, not with regard to the methods, but concerning materials. This is because the description (3.3) is not sufficient for a possible reproduction. Data sources are not well explained, also some variable is not clear.
For example:
- rows 235-237 “xij represents the input variable matrix, and the specific indicators include planting area, fertilizer, pesticide, agricultural film, diesel oil, seed, electricity for irrigation, labor and machinery”. How fertilizer or pesticide are measured (tons? Expenses?) For agriculture as a whole or only for maize?
- rows 302-303 “Grain production capacity (GPC): GPCit= GPit / APit (10) where GPit represents the grain production; APit represents the agricultural practitioner”. Ok for grain production, but exactly means agricultural practitioner, how it is measured?
And so on
The article is good, but it needs some adjustments, perhaps an appendix in which data, sources and variables are clearly described.
Author Response

(The authors gave the same response as above.)

Reviewer 3 Report
The scientific article solves a topical problem.
Its presentation, from literature review to completion, is well presented and logically structured.
The conclusions are profound and reasonable.
The only part that could be deepened is the broader justification of the chosen analytical model.
The article is suitable for publication.
Author Response

(The authors gave the same response as above.)
